# Antihypertensive Drugs and the Risk of Cancer: A Nationwide Cohort Study

**DOI:** 10.3390/jcm10040771

**Published:** 2021-02-15

**Authors:** In-Jeong Cho, Jeong-Hun Shin, Mi-Hyang Jung, Chae Young Kang, Jinseub Hwang, Chang Hee Kwon, Woohyeun Kim, Dae-Hee Kim, Chan Joo Lee, Si-Hyuck Kang, Ju-Hee Lee, Hack-Lyoung Kim, Hyue Mee Kim, Iksung Cho, Hae-Young Lee, Wook-Jin Chung, Sang-Hyun Ihm, Kwang Il Kim, Eun Joo Cho, Il-Suk Sohn, Sungha Park, Jinho Shin, Sung Kee Ryu, Jang Young Kim, Seok-Min Kang, Myeong-Chan Cho, Wook Bum Pyun, Ki-Chul Sung

**Affiliations:** 1Division of Cardiology, Department of Internal Medicine, Ewha Womans University Seoul Hospital, Ewha Womans University College of Medicine, Seoul 07985, Korea; injeong.md@gmail.com; 2Division of Cardiology, Department of Internal Medicine, Hanyang University College of Medicine, Seoul 04763, Korea; cardio.hyapex@gmail.com (J.-H.S.); coincidence1@naver.com (W.K.); jhs2003@hanyang.ac.kr (J.S.); 3Cardiovascular Center, Dongtan Sacred Heart Hospital, Hallym University College of Medicine, Hwaseong 18450, Korea; floria0515@gmail.com; 4Department of Statistics and Computer Science, Daegu University, Gyeongsan-si 38453, Korea; codud12311@naver.com (C.Y.K.); hjs04090409@gmail.com (J.H.); 5Division of Cardiology, Department of Internal Medicine, Konkuk University Medical Center, Konkuk University School of Medicine, Seoul 05030, Korea; vertex_77@naver.com; 6Division of Cardiology, Asan Medical Center, University of Ulsan College of Medicine, Seoul 05505, Korea; daehee74@amc.seoul.kr; 7Division of Cardiology, Severance Cardiovascular Hospital and Cardiovascular Research Institute, Yonsei University College of Medicine, Seoul 03722, Korea; zanzu@yuhs.ac (C.J.L.); iksungcho.md@gmail.com (I.C.); shpark0530@yuhs.ac (S.P.); SMKANG@hyuh.ac (S.-M.K.); 8Department of Internal Medicine, Seoul National University Bundang Hospital, Seoul National University College of Medicine, Seongnam-si 13620, Korea; eandp303@snu.ac.kr (S.-H.K.); kikim907@snu.ac.kr (K.I.K.); 9Division of Cardiology, Department of Internal Medicine, Chungbuk National University Hospital, Chungbuk National University College of Medicine, Cheongju-si 28644, Korea; juheelee@chungbuk.ac.kr (J.-H.L.); mccho@cbnu.ac.kr (M.-C.C.); 10Department of Internal Medicine, Boramae Medical Center, Seoul National University College of Medicine, Seoul 07061, Korea; khl2876@gmail.com; 11Division of Cardiology, Department of Internal Medicine, Chung-Ang University Hospital, Seoul 06973, Korea; chemie27@naver.com; 12Division of Cardiology, Department of Internal Medicine, Seoul National University Hospital, Seoul 03080, Korea; hylee612@snu.ac.kr; 13Department of Cardiovascular Medicine, Gachon University Gil Medical Center, Incheon 21565, Korea; wjcheart@gmail.com; 14Division of Cardiology, Department of Internal Medicine, Bucheon St. Mary’s Hospital, The Catholic University of Korea, Bucheon-si 14647, Korea; limsh@catholic.ac.kr; 15Division of Cardiology, Department of Internal Medicine, Yeouido St. Mary’s Hospital, The Catholic University of Korea, Seoul 07345, Korea; choej4oct@gmail.com; 16Division of Cardiology, Department of Internal Medicine, Kyung Hee University at Gangdong, Seoul 05278, Korea; issohn89@hanmail.net; 17Division of Cardiology, Department of Internal Medicine, Eulji Medical School of Medicine, Seoul 01830, Korea; ysk1140@eulji.ac.kr; 18Division of Cardiology, Department of Internal Medicine, Wonju College of Medicine, Yonsei University, Wonju-si 26426, Korea; kimjang713@gmail.com; 19Division of Cardiology, Department of Internal Medicine, Kangbuk Samsung Hospital, Sungkyunkwan University School of Medicine, Seoul 03181, Korea

**Keywords:** antihypertensive agent, neoplasms, cohort study

## Abstract

We sought to assess the association between common antihypertensive drugs and the risk of incident cancer in treated hypertensive patients. Using the Korean National Health Insurance Service database, the risk of cancer incidence was analyzed in patients with hypertension who were initially free of cancer and used the following antihypertensive drug classes: Angiotensin-converting enzyme inhibitors (ACEIs); angiotensin receptor blockers (ARBs); beta blockers (BBs); calcium channel blockers (CCBs); and diuretics. During a median follow-up of 8.6 years, there were 4513 (6.4%) overall cancer incidences from an initial 70,549 individuals taking antihypertensive drugs. ARB use was associated with a decreased risk for overall cancer in a crude model (hazard ratio (HR): 0.744, 95% confidence interval (CI): 0.696–0.794) and a fully adjusted model (HR: 0.833, 95% CI: 0.775–0.896) compared with individuals not taking ARBs. Other antihypertensive drugs, including ACEIs, CCBs, BBs, and diuretics, did not show significant associations with incident cancer overall. The long-term use of ARBs was significantly associated with a reduced risk of incident cancer over time. The users of common antihypertensive medications were not associated with an increased risk of cancer overall compared to users of other classes of antihypertensive drugs. ARB use was independently associated with a decreased risk of cancer overall compared to other antihypertensive drugs.

## 1. Introduction

Treatment for hypertension has been shown to decrease cardiovascular morbidity and mortality [1,2]. The 2017 American College of Cardiology/American Heart Association guidelines for high blood pressure for adults recommend diuretics, angiotensin-converting enzyme inhibitors (ACEIs), angiotensin receptor blockers (ARBs), and calcium channel blockers (CCBs) as primary agents and beta blockers (BBs) and alpha blockers as secondary agents for hypertension treatment in cases without compelling indications, such as heart failure, kidney disease, and angina [1]. A variable proportion of patients with hypertension are treated with antihypertensive drugs, including ACEIs, ARBs, BBs, CCBs, and diuretics in different countries [3]. Given that antihypertensive drugs are used long term, without discontinuation, and that cancer is the second leading cause of mortality worldwide [4], it is reasonable to investigate incident cancer as a possible side effect when selecting the primary agents for hypertension treatment [5].

Whether antihypertensive medications contribute to the risk of cancer remains unclear, with variable data according to cancer type and antihypertensive drug class [6,7,8,9,10]. Furthermore, numerous data have shown an association between hypertension and cancer incidence, especially for kidney cancer [11,12], breast cancer [13], and colorectal cancer [14], which makes the potential association between antihypertensive drugs and cancer more complex. ARBs, which represent the most recently developed antihypertensive drug class, were introduced in the 1990s [15], and the long-term use of ARBs in relation to incident cancer still requires further investigation. The potential for the cancer risk of ARBs was first raised in the Candesartan trial [16], which followed multiple studies with conflicting results [6,17,18,19,20]. ARBs were associated with an increased risk of cancer development in a meta-analysis [17], while two other subsequent meta-analyses did not find an excess cancer risk among ARB users compared with controls [6,18]. Furthermore, several mechanistic pieces of evidence provide a biological rationale for the possible antitumor effect of ARBs via angiotensin-type 2 receptor (AT2R) stimulation [21,22,23]. Therefore, this study aimed to assess the association between antihypertensive medication use and the risk of incident cancer in treated hypertensive patients using a large nationwide cohort in Korea.

## 2. Methods

### 2.1. Study Population

Patients who were diagnosed with essential hypertension were identified based on International Classification of Diseases, 10th revision (ICD-10) codes I10-13 between 1 January 2005 and 31 December 2012 in the National Health Insurance System (NHIS) database in Korea. Hospitalizations ≥1 or outpatient visits ≥2 with the above diagnoses were defined as hypertension. Among the 2,140,096 candidates with hypertension who we initially screened, we selected 537,095 patients without missing health screening examination data, as detailed clinical information, laboratory tests, and the socioeconomic status were available in the health screening examination database. To verify the new cancer incidence, 59,503 patients with cancer diagnoses established before the baseline date were excluded. From 477,592 hypertensive patients without baseline cancer, we excluded 24,489 individuals who were not prescribed hypertension drugs, 249,504 patients who were prescribed hypertension drugs for less than 1 year, and 123,050 who were using antihypertension drug at the baseline. As the duration of drug use and whether the patient had taken other antihypertensive drugs before the index period could not be assessed, we excluded prevalent users and those who used antihypertensive drugs at the baseline. Finally, 70,549 patients who initiated hypertension medication during the study period and continued taking the drug for ≥1 year were included in the study population. Details of the selection of the study population are shown in Figure 1. All participants were followed from the baseline to the date of cancer diagnosis (event of interest), death by any reason (competing event), censoring due to a loss of health insurance eligibility, or the end of the follow-up period (31 December 2017), whichever occurred first.

### 2.2. Definition of Cancer

In the healthcare utilization database, participants with overall cancer during follow-up were identified using ICD-10 codes. Hospitalizations ≥ 1 with ICD-10 codes C00–C96 were defined as an incidence of overall cancer. Site-specific cancers were defined as follows, according to ICD-10 codes: Lung, C34; colorectal, C18-20; breast, C50; prostate, C61; bladder, C67; pancreatic, C25; kidney, C64; hepatic, C22; and gastric cancer, C16 (Appendix A).

### 2.3. Assessment of Antihypertensive Drug Use

For the antihypertensive medication exposure assessment, we employed healthcare utilization data collected between 1 January 2005 and 31 December 2012. Prescription data on the following five antihypertensive medications were assessed based on anatomic therapeutic chemical codes: ACEIs; ARBs; BBs; CCBs; and diuretics. Participants who were prescribed antihypertensive medications for at least 1 year were defined as medication users. Non-users were subjects who had never used the medication.

### 2.4. Other Covariates

For information on health-related lifestyle factors, such as the smoking status and alcohol consumption, body mass index (BMI), and systolic blood pressure (SBP), we used the most recent NHIS health screening data from the time of hypertension diagnosis. Data on diagnoses of diabetes (E10–13), heart failure (I509), and chronic obstructive pulmonary disease (J449) were collected using ICD-10 codes (Appendix A).

### 2.5. Data Source

This nationwide retrospective cohort study used data from the NHIS in Korea collected from 1 January 2005 to 31 December 2017. The National Health Insurance Service is a single insurance provider in Korea and covers 97% of the Korean population. The NHIS claim database includes information regarding demographic characteristics, diagnoses, prescriptions, death, and health screening examination data. The NHIS health screening examination data include health questionnaires and laboratory tests. This study was approved by the Institutional Review Board of Kangbuk Samsung Hospital (KBSMC 2019-12-021). Data were fully anonymized for all analyses, and the need for informed consent was waived. The authors are restricted from sharing the data used for this study because the Korean NHIS owns the data. This study was performed as a project between the Korean Society of Hypertension and the NHIS. Researchers who are not members of the collaboration can request access to the NHIS website http://nhiss.nhis.or.kr (accessed on 2 July 2020). Details of this process and a provision guide are now available at https://nhiss.nhis.or.kr/bd/ab/bdaba032eng.do (accessed on 2 July 2020).

### 2.6. Statistical Analysis

Incidence rates were estimated using the total number of outcomes during follow-up divided by 1000 person-years. The competing risk was analyzed with the Fine and Gray model, which was performed by considering death as a competing event [24]. The association of each antihypertensive medication (ACEI, ARB, BB, CCB, and diuretics) with the risk of incident cancer was evaluated using three models: A crude model; an age- and sex-adjusted model; and a fully adjusted model. The fully adjusted model used the possible confounders of age; sex; BMI; SBP; the alcohol consumption frequency; the income status; comorbidities, including diabetes, heart failure, and chronic obstructive pulmonary disease; and the use of each antihypertensive medication. Age, BMI, and SBP were continuous variables and the others were categorical variables. There were no statistically significant interactions between antihypertensive medication usage, and the usage of other drugs served as a confounder for regression analysis to determine the risk of incident cancer for each antihypertensive medication. Fine and Gray competing risk analysis models reporting hazard ratios (HRs) and 95% confidence intervals (CIs) were employed to determine potentially useful variables for predicting cancer occurrence. Long-term risk association between antihypertensive drug use and incident cancer was analyzed using lag-time analysis, where medication use was lagged 1, 2, and 3 years forward in follow-up time, that is, relating medication use to the subsequent cancer risk 1–3 years later. A *p*-value < 0.05 was considered statistically significant.

## 3. Results

### 3.1. Study Population

Table 1 demonstrates the baseline characteristics of the study population. Of the 70,549 hypertensive patients analyzed, there were 4210 ACEI users, 55,645 ARB users, 13,158 BB users, 51,036 CCB users, and 32,990 diuretic users. There was an overall cancer incidence of 4513 (6.4%) during a median follow-up of 8.6 years (interquartile range (IQR): 6.7–10.8 years). The median drug exposure duration was 3.3 years (IQR: 1.8–5.6 years) for ACEI users, 5.5 years (IQR: 3.5–7.2 years) for ARB users, 3.8 years (IQR: 2.1–6.4 years) for BB users, 5.3 years (IQR: 3.0–7.6 years) for CCB users, and 4.1 years (IQR: 2.2–6.2 years) for diuretic users. The mean age at screening was 55.5 ± 9.2 years, and 60.9% of the analyzed population was male. A comparison of baseline characteristics between individuals with and without incident cancer during the follow-up period is shown in Appendix A.

### 3.2. Cancer Risk by Antihypertensive Medication

Table 2 shows the overall cancer risk by antihypertensive drug use. ARB use was associated with a decreased risk for overall cancer compared with ARB non-users in a crude model (HR: 0.744, 95% CI 0.696–0.794), an age- and sex-adjusted model (HR: 0.822, 95% CI: 0.769–0.878), and a fully adjusted model (HR: 0.833, 95% CI: 0.775–0.896). Other antihypertensive drugs, including ACEI, CCB, BB, and diuretics, did not show significant associations with the cancer risk in fully adjusted models.

To analyze the possibility of the time-lag effect of antihypertensive use on the incidence of cancer, we used HR with lag times. Figure 2 shows the cancer risk by the time lags of 1 year, 2 years, and 3 years in individuals on ARBs. ARB use was associated with a decreased cancer risk in analyses with a 1-year lag time (HR: 0.619, 95% CI: 0.574–0.668), 2-year lag time (HR: 0.536, 95% CI: 0.496–0.579), and 3-year lag time (HR: 0.445, 95% CI: 0.41–0.483).

Table 3 demonstrates the site-specific cancer risk by antihypertensive drug use. The use of ARBs, CCBs, BBs, and diuretics showed a decreased cancer risk for several specific organs. ARBs were associated with a decreased risk for hepatic cancer (HR: 0.665, 95% CI: 0.519–0.851) and gastric cancer (HR: 0.759, 95% CI: 0.636–0.906). BBs were associated with a decreased cancer risk for pancreatic cancer (HR: 0.447, 95% CI: 0.243–0.822). CCBs were associated with a decreased risk for gastric cancer (HR: 0.823, 95% CI: 0.694–0.976), and diuretics were associated with a decreased risk for breast cancer (HR: 0.747, 95% CI: 0.575–0.972). Among the various cancers, an increased cancer risk was only shown for bladder cancer, for which the cancer risk was increased in BB users compared to non-users (HR: 1.505, 95% CI: 1.074–2.108).

### 3.3. Risk Trends by Duration of ARB Use and Subgroups

There were 3294 (5.9%) overall cancer incidents during follow-up in ARB users. Figure 3 shows the overall cancer risk trends by duration of ARB use. The long-term use of ARBs was significantly associated with a reduced risk of overall cancer. The adjusted HR (95% CI) was 0.907 (0.828–0.993) for 3–4 years of use, 0.707 (0.645–0.775) for 5–6 years of use, 0.698 (0.632–0.772) for 7–8 years of use, and 0.670 (0.589–0.761) for ≥9 years of use, suggesting that the decrease in the risk of incident cancer overall was greater with a longer duration of ARB use.

Subgroup analysis according to age, sex, BMI, SBP, the alcohol consumption frequency, diabetes, and SBP is shown in Appendix A. There were no statistically significant interactions between ARB use and factors such as age, the smoking status, the presence of diabetes, BMI, and the baseline SBP. Being male was associated with a greater decrease in overall cancer incidence with ARB use compared to being female (HR: 0.819, 95% CI: 0.784–0.897 for men and HR: 0.883, 95% CI: 0.783–0.996 for women; *p*-value for interaction = 0.003). Those with the highest alcohol consumption showed the greatest risk reduction compared to others (HR: 0.663, 95% CI: 0.512–0.858 for alcohol consumption frequency ≥5 times per week, *p*-value for interaction = 0.003).

## 4. Discussion

The principal finding of the current study was that the use of common antihypertensive drugs was not associated with an increased risk of cancer overall compared to the use of other hypertensive drugs. Among five common antihypertensive drugs (ACEI, BB, CCB, ARB, and diuretics), ARB use was independently associated with a decreased risk of cancer overall compared to other antihypertensive drugs, and the risk of incident cancer overall was decreased further with a longer duration of ARB use.

The strengths of our study include the large, nationwide sample with the virtually complete capture of cancer cases, since we used the NHIS claims database and included individuals who had completed NHIS health screening examinations. The long follow-up period (median, 8.6 years) is another strength of our study. The NHIS in Korea provides insurance benefits and free health screening programs, including cancer screening, for all citizens of Korea, and all adults over 40 years old are recommended to undergo periodic health examinations. The participation rate was as high as 74.8% in 2014 [25]. Therefore, our study population represents a general hypertensive population with a complete survey for incident cancer. To clarify causal associations, we performed duration-relation analysis and found that the protective effect was pronounced in patients with longer durations of ARB use (3 or more years). This strong duration-relation possibly supports the causal association of ARB use with a decrease in the incidence of cancer. Furthermore, we assessed the long-term risk association between antihypertensive drug use and incident cancer using lag-time analysis, and showed that ARB use was associated with a decreased subsequent overall cancer risk 1–3 years later, which also suggests a causal relationship between ARB use and incident cancer overall.

ARBs were developed as antihypertensive agents, but a series of major trials have demonstrated the benefits of ARBs across a spectrum of cardiovascular and renal diseases, and even improvement in glycemic control [26]. In addition to the previously reported favorable cardiovascular and metabolic impacts of ARBs, we found that ARBs were associated with a decreased overall risk of cancer of about 17% and that the risk of incident cancer overall decreased with a longer duration of ARB use, up to 33% in individuals with ≥9 years of use. Previous research on ARB use has yielded conflicting results regarding the cancer risk [6,17,19,27,28,29]. The pathogenesis of ARB in cancer development has not yet been established. ARB inhibits the renin-angiotensin-aldosterone system selectively at the angiotensin-type 1 receptor (AT1R) and leaves the AT2R open for stimulation. Angiotensin II may play a role in cell growth and proliferation and angiogenesis, mainly through AT1R signaling. The long-term antagonism of AT1R by ARB may result in the persistent activation of AT2R signaling, the role of which has not yet been established in cancer [30]. However, several investigators have suggested that AT2R stimulation results in an antitumor effect [21,22,23].

From site-specific analysis, we speculated that the reduced cancer risk associated with ARB use is driven by a decrease in hepatic cancer and gastric cancer, although our analysis per type of cancer should be interpreted with caution and warrants further investigation. Theoretically, neoangiogenesis is one of the key pathogenic mechanisms in hepatic cancer, and modulation of the renin-angiotensin-aldosterone system seems to be a possible anti-tumor mechanism of the anti-angiogenic and anti-fibrogenic activity of ARBs [31]. AT1R is upregulated in human gastric cancer and may be involved in the progression of gastric cancer. In addition, the growth of gastric cancer cells was significantly suppressed by treatment with an AT1R antagonist [32]. Further studies are warranted to confirm whether this reduced cancer risk is solely associated with these site-specific cancers or whether it affects all cancers through interruption of the angiogenesis of tumor cells by AT1R antagonists and the antitumor effects of AT2R stimulation [33].

Hypertension and cancer are two major public health concerns worldwide. An association between hypertension and cancer incidence has been reported, especially for kidney cancer [11,12], breast cancer [13], and colorectal cancer [14], but these findings remain controversial. Given that antihypertensive drugs are used over long-term periods without discontinuation, the risk of cancer should be carefully considered when selecting the primary agent for hypertension treatment. Therefore, it is notable that no common antihypertensive drugs increased the risk of incident cancer overall compared to other drugs in our study in treated hypertensive patients. Interestingly, the risk of kidney cancer was reported to be significantly increased with a higher blood pressure in a dose-dependent manner, even after adjustment for antihypertensive drug use in a Korean cohort [12]. As our study population demonstrated similar SBP levels across different antihypertensive classes and SBP levels during follow-up were not determined, the impact of blood pressure, rather than antihypertensive drug classes, on cancer incidence remains unclear. Further studies are needed to investigate the association between the blood pressure control status and cancer incidence in different antihypertensive drug classes.

We found that the incidence of overall cancer was significantly reduced in ARB users compared to other drugs, even in the case of long-term usage of ≥9 years, which supports the long-term safety of antihypertensive drugs, especially ARBs. Our study demonstrated that no specific antihypertensive drugs were generally associated with an increase in cancer risk, as evidenced by the fact that the incidence of several site-specific cancers was also reduced in BB, CCB, and diuretic users, in addition to ARB users. The incident cancer risk was only increased in bladder cancer for BB users, but larger studies are needed to clarify whether this association exists, since the small number of bladder cancer incidents in our study limited further detailed analysis.

### Limitations

This study had several limitations. First of all, this study is a retrospective, non-randomized, cohort study analysis, which can lead to selection bias. In general, this cohort represented a high-income and relatively young Korean population with a short period of antihypertensive drug use. Therefore, it may not be possible to generalize these results to a broader population with a longer median duration of antihypertensive drug use. However, this study is based on a large-scale, nationwide, population-based database that provides the avoidance of selection bias. Moreover, we performed adjustments for several confounders and conducted competing risk analyses to reduce bias. Second, the diagnosis in NHIS primarily serves the purpose of administrative billing and was not originally intended to perform statistical analysis. However, data on the prescription of antihypertensive drugs and cancer diagnoses were very reliable. Third, we did not have data regarding the dosage of antihypertensive drugs, which did not permit an analysis of the association between the cumulative dose and cancer incidence, which might have strengthened our hypothesis. Fourth, the study population of this study only included antihypertensive drug users; a decreased cancer incidence risk in ARB users is expressed relative to users of other antihypertensive drugs and not compared to the general population. Further analysis of a comparison of users of antihypertensive drugs to those not using antihypertensive drugs would be needed to reduce bias. Fifth, the prevalence of ACEI users was very low compared with those taking other antihypertensive drugs, which might have affected the statistical results. Sixth, since we did not incorporate the antihypertensive drug usage as a time-dependent variable, immortal time bias could have occurred, although individuals enrolled in this study were taking at least one antihypertensive drug. Seventh, the control of drug adherence was lacking due to the retrospective nature of this study. Therefore, further studies are warranted to validate the results of the current study with a longer duration follow-up. Additional analysis evaluating the risk of incident cancer among subjects with the simultaneous use of two or more different classes of antihypertensive drugs would also provide useful insights.

## 5. Conclusions

The use of common antihypertensive medications was not associated with an increased risk of cancer overall compared to the use of other classes of antihypertensive drugs. ARB use was independently associated with a decreased risk of cancer overall compared to other antihypertensive drugs.

## Figures and Tables

**Figure 1 jcm-10-00771-f001:**
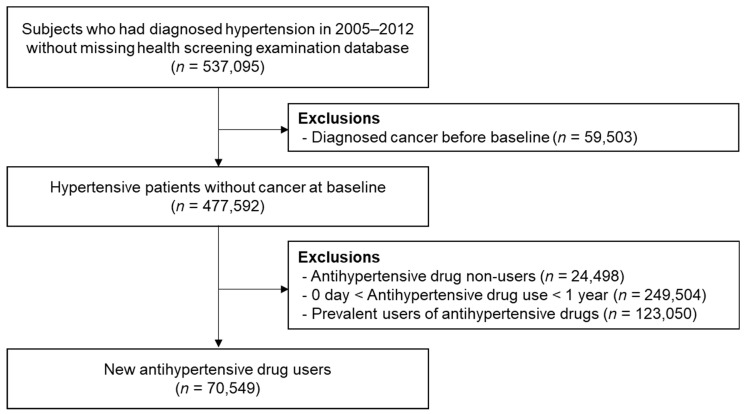
Selection of the study population.

**Figure 2 jcm-10-00771-f002:**
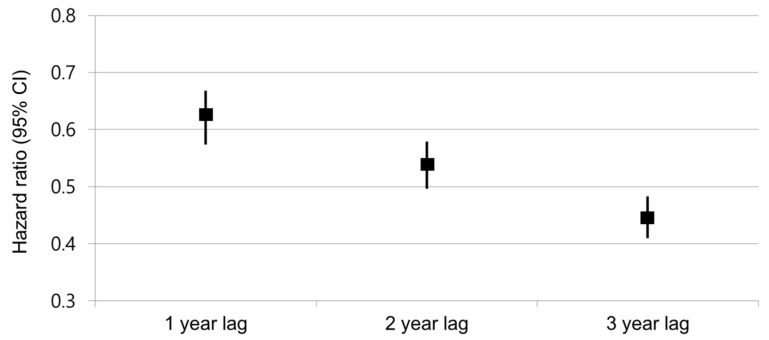
Cancer risk by the time lags of angiotensin receptor blocker use. Compared with non-users of ARBs. CI, confidence interval. Square boxes indicate hazard ratios and vertical bars indicate 95% confidence intervals.

**Figure 3 jcm-10-00771-f003:**
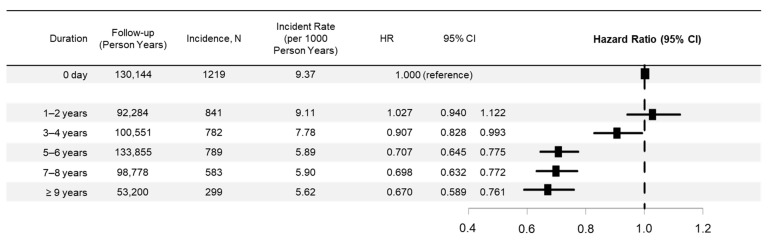
Cancer risk trends by the duration of angiotensin receptor blocker use. HR, hazard ratio and CI, confidence interval. Square boxes indicate hazard ratios and horizontal bars indicate 95% confidence intervals.

**Table 1 jcm-10-00771-t001:** Baseline characteristics.

Variable	All Users	ACEI	ARB	BB	CCB	Diuretics
Number, N	70,549	4210	55,645	13,158	51,036	32,990
Number of overall cancers, N (%)	4513 (6.4)	362 (8.6)	3294 (5.9)	982 (7.5)	3443 (6.8)	2155 (6.5)
Follow-up, years	8.6 (6.7–10.8)	10.7 (8.4–12.0)	8.5 (6.7–10.7)	10.0 (7.7–11.8)	8.8 (6.9–11.0)	9.1 (7.2–11.3)
Drug exposure duration, years	-	3.3 (1.8–5.6)	5.5 (3.5–7.2)	3.8 (2.1–6.4)	5.3 (3.0–7.6)	4.1 (2.2–6.2)
Deaths, N (%)	3025 (4.3)	353 (8.4)	2050 (3.7)	741 (5.6)	2333 (4.6)	1542 (4.7)
Age, years	55.2 ± 9.2	57.0 ± 9.5	54.5 ± 9.0	55.4 ± 9.2	55.4 ± 9.2	55.4 ± 9.3
Male sex, N (%)	42,990 (60.9)	3001 (71.3)	35,113 (63.1)	8270 (62.9)	31,668 (62.1)	19,747 (59.9)
BMI, kg/m^2^	24.9 ± 3.0	24.7 ± 3.0	25.1 ± 3.0	25.0 ± 3.1	25.0 ± 3.0	25.2 ± 3.1
SBP, mmHg	139.7 ± 18.3	138.2 ± 19.6	140.5 ± 18.4	141.5 ± 21.1	141.8 ± 18.5	142.3 ± 19.0
Smoking, N (%)						
Current	16,214 (23.0)	1053 (25.0)	13,525 (24.3)	3249 (24.7)	12,277 (24.1)	7882 (23.9)
Past smoker	12,155 (17.2)	712 (16.9)	9872 (17.7)	1979 (15.0)	8509 (16.7)	5050 (15.3)
Never-smoker	42,180 (59.8)	2445 (58.1)	32,248 (58.0)	7930 (60.3)	30,250 (59.3)	20,058 (60.8)
Alcohol, N (%)						
None	35,575 (50.4)	2181 (51.8)	26,871 (48.3)	6759 (51.4)	24,965 (48.9)	16,387 (49.7)
1–2/week	22,265 (31.6)	1323 (31.4)	18,146 (32.6)	4109 (31.2)	16,322 (32.0)	10,370 (31.4)
3–4/week	8302 (11.8)	431 (10.2)	6969 (12.5)	1484 (11.3)	6308 (12.4)	3978 (12.1)
≥5/week	4407 (6.3)	275 (6.5)	3659 (6.6)	806 (6.1)	3441 (6.7)	2255 (6.8)
Income, N (%)						
Low (1st-3rd deciles)	17,221 (24.4)	1000 (23.8)	13,456 (24.2)	3374 (25.6)	12,635 (24.8)	8444 (25.6)
Middle (4th-7th deciles)	23,982 (34.0)	1458 (34.6)	19,006 (34.2)	4596 (34.9)	17,676 (34.6)	11,720 (35.5)
High (8th-10th deciles)	29,346 (41.6)	1752 (41.6)	23,183 (41.7)	5188 (39.4)	20,725 (40.6)	12,826 (38.9)
Comorbidities						
Diabetes, N (%)	13,382 (19.0)	1156 (27.5)	10,854 (19.5)	1996 (15.2)	8172 (16.0)	5573 (16.9)
Heart failure, N (%)	216 (0.3)	29 (0.7)	154 (0.3)	57 (0.4)	113 (0.2)	86 (0.3)
COPD, N (%)	1704 (2.4)	102 (2.4)	1227 (2.2)	270 (2.1)	1127 (2.2)	752 (2.3)

ACEI, angiotensin converting enzyme inhibitor; ARB, angiotensin receptor blocker; BB, beta blocker; CCB, calcium channel blocker; BMI, body mass index; SBP, systolic blood pressure; COPD, chronic obstructive pulmonary disease.

**Table 2 jcm-10-00771-t002:** Overall cancer risk by antihypertensive drug use.

Variable		Follow-Up (Person Years)	Cancer Incidence, N	Incident Rate (per 1000 Person Years)	Unadjusted		Age- and Sex-Adjusted		Fully Adjusted	
					HR	95% CI	HR	95% CI	HR	95% CI
ACEI	No	566,817	4151	7.32	1.000 (reference)		1.000 (reference)		1.000 (reference)	
	Yes	41,995	362	8.62	1.148	1.031–1.277	1.022	0.918–1.138	1.016	0.912–1.132
ARB	No	130,144	1219	9.37	1.000 (reference)		1.000 (reference)		1.000 (reference)	
	Yes	478,668	3294	6.88	0.744	0.696–0.794	0.822	0.769–0.878	0.833	0.775–0.896
BB	No	483,462	3531	7.30	1.000 (reference)		1.000 (reference)		1.000 (reference)	
	Yes	125,350	982	7.83	1.055	0.983–1.132	1.033	0.962–1.109	1.03	0.958–1.107
CCB	No	158,876	1070	6.73	1.000 (reference)		1.000 (reference)		1.000 (reference)	
	Yes	449,937	3443	7.65	1.124	1.049–1.204	1.081	1.009–1.158	1.053	0.981–1.13
Diuretics	No	309,741	2358	7.61	1.000 (reference)		1.000 (reference)		1.000 (reference)	
	Yes	299,071	2155	7.21	0.936	0.883–0.992	0.92	0.868–0.976	0.957	0.898–1.019

ACEI, angiotensin converting enzyme inhibitor; ARB, angiotensin receptor blocker; BB, beta blocker; CCB, calcium channel blocker; HR, hazard ratio; CI, confidence interval.

**Table 3 jcm-10-00771-t003:** Site-specific cancer risk by antihypertensive drug use.

Variable		Follow-Up (Person Years)	Cancer Incidence, N	Incident Rate (per 1000 Person Years)	Unadjusted		Age- and Sex-Adjusted		Fully Adjusted	
					HR	95% CI	HR	95% CI	HR	95% CI
Lung cancer										
ACEI	No	581,613	509	0.88	1.000 (reference)		1.000 (reference)		1.000 (reference)	
	Yes	43,359	54	1.25	1.335	1.009–1.768	1.019	0.768–1.351	0.998	0.752–1.326
ARB	No	134,482	159	1.18	1.000 (reference)		1.000 (reference)		1.000 (reference)	
	Yes	490,490	404	0.82	0.716	0.595–0.861	0.843	0.699–1.015	0.86	0.705–1.049
BB	No	495,930	433	0.87	1.000 (reference)		1.000 (reference)		1.000 (reference)	
	Yes	129,042	130	1.01	1.1	0.904–1.339	1.046	0.859–1.274	1.053	0.861–1.288
CCB	No	162,677	117	0.72	1.000 (reference)		1.000 (reference)		1.000 (reference)	
	Yes	462,295	446	0.96	1.298	1.06–1.59	1.186	0.968–1.453	1.186	0.965–1.458
Diuretics	No	318,319	283	0.89	1.000 (reference)		1.000 (reference)		1.000 (reference)	
	Yes	306,652	280	0.91	0.993	0.843–1.171	0.964	0.817–1.137	1.019	0.853–1.216
Colorectal cancer										
ACEI	No	580,493	589	1.01	1.000 (reference)		1.000 (reference)		1.000 (reference)	
	Yes	43,172	61	1.41	1.332	1.021–1.737	1.105	0.847–1.442	1.118	0.858–1.456
ARB	No	134,149	176	1.31	1.000 (reference)		1.000 (reference)		1.000 (reference)	
	Yes	489,516	474	0.97	0.755	0.635–0.898	0.868	0.728–1.036	0.857	0.711–1.031
BB	No	494,933	502	1.01	1.000 (reference)		1.000 (reference)		1.000 (reference)	
	Yes	128,732	148	1.15	1.098	0.914–1.2318	1.062	0.884–1.276	1.029	0.856–1.236
CCB	No	162,441	133	0.82	1.000 (reference)		1.000 (reference)		1.000 (reference)	
	Yes	461,224	517	1.12	1.342	1.108–1.624	1.258	1.039–1.524	1.178	0.968–1.432
Diuretics	No	317,700	319	1.00	1.000 (reference)		1.000 (reference)		1.000 (reference)	
	Yes	305,965	331	1.08	1.056	0.906–1.23	1.033	0.886–1.204	1.031	0.877–1.213
Hepatic cancer										
ACEI	No	581,886	346	0.59	1.000 (reference)		1.000 (reference)		1.000 (reference)	
	Yes	43,403	28	0.65	1.029	0.704–1.504	0.833	0.568–1.223	0.795	0.542–1.605
ARB	No	134,553	108	0.80	1.000 (reference)		1.000 (reference)		1.000 (reference)	
	Yes	490,736	266	0.54	0.69	0.551–0.864	0.692	0.552–0.868	0.665	0.519–0.851
BB	No	496,177	292	0.59	1.000 (reference)		1.000 (reference)		1.000 (reference)	
	Yes	129,111	82	0.64	1.042	0.817–1.329	0.995	0.78–1.269	0.992	0.776–1.269
CCB	No	162,754	79	0.49	1.000 (reference)		1.000 (reference)		1.000 (reference)	
	Yes	462,535	295	0.64	1.286	1.002–1.65	1.205	0.939–1.547	1.184	0.918–1.528
Diuretics	No	318,441	197	0.62	1.000 (reference)		1.000 (reference)		1.000 (reference)	
	Yes	306,848	177	0.58	0.911	0.745–1.114	0.916	0.748–1.121	1.02	0.818–1.271
Gastric cancer										
ACEI	No	580,103	681	1.17	1.000 (reference)		1.000 (reference)		1.000 (reference)	
	Yes	43,157	74	1.71	1.427	1.123–1.812	1.151	0.904–1.466	1.16	0.91–1.479
ARB	No	133,969	217	1.62	1.000 (reference)		1.000 (reference)		1.000 (reference)	
	Yes	489,291	538	1.10	0.691	0.59–0.81	0.763	0.65–1.895	0.759	0.636–0.906
BB	No	494,612	597	1.21	1.000 (reference)		1.000 (reference)		1.000 (reference)	
	Yes	128,648	158	1.23	1	0.839–1.192	0.96	0.805–1.143	0.937	0.784–1.119
CCB	No	162,142	200	1.23	1.000 (reference)		1.000 (reference)		1.000 (reference)	
	Yes	461,118	555	1.20	0.963	0.819–1.132	0.897	0.763–1.055	0.823	0.694–0.976
Diuretics	No	317,323	407	1.28	1.000 (reference)		1.000 (reference)		1.000 (reference)	
	Yes	305,937	348	1.14	0.877	0.76–1.011	0.865	0.75–0.998	0.904	0.772–1.059
Bladder cancer										
ACEI	No	582,117	155	0.27	1.000 (reference)		1.000 (reference)		1.000 (reference)	
	Yes	43,386	18	0.41	1.49	0.914–2.429	1.096	0.671–1.79	1.078	0.65–1.788
ARB	No	134,664	46	0.34	1.000 (reference)		1.000 (reference)		1.000 (reference)	
	Yes	490,839	127	0.26	0.774	0.553–1.084	0.884	0.629–1.244	0.996	0.693–1.433
BB	No	496,417	122	0.25	1.000 (reference)		1.000 (reference)		1.000 (reference)	
	Yes	129,087	51	0.40	1.564	1.123–2.179	1.474	1.057–2.054	1.505	1.074–2.108
CCB	No	162,790	37	0.23	1.000 (reference)		1.000 (reference)		1.000 (reference)	
	Yes	462,714	136	0.29	1.269	0.882–1.826	1.151	0.8–1.657	1.135	0.78–1.651
Diuretics	No	318,566	95	0.30	1.000 (reference)		1.000 (reference)		1.000 (reference)	
	Yes	306,937	78	0.25	0.836	0.618–1.13	0.819	0.606–1.109	0.781	0.562–1.085
Breast cancer										
ACEI	No	581,459	267	0.46	1.000 (reference)		1.000 (reference)		1.000 (reference)	
	Yes	43,407	8	0.18	0.413	0.204–0.836	0.599	0.296–1.212	0.609	0.299–1.24
ARB	No	134,451	71	0.53	1.000 (reference)		1.000 (reference)		1.000 (reference)	
	Yes	490,416	204	0.42	0.792	0.604–1.038	0.936	0.713–1.229	1.05	0.777–1.418
BB	No	495,860	216	0.44	1.000 (reference)		1.000 (reference)		1.000 (reference)	
	Yes	129,007	59	0.46	1.08	0.809–1.442	1.154	0.865–1.541	1.203	0.895–1.617
CCB	No	162,611	67	0.41	1.000 (reference)		1.000 (reference)		1.000 (reference)	
	Yes	462,256	208	0.45	1.114	0.884–1.468	1.263	0.957–1.667	1.194	0.896–1.593
Diuretics	No	318,193	154	0.48	1.000 (reference)		1.000 (reference)		1.000 (reference)	
	Yes	306,674	121	0.39	0.834	0.657–1.058	0.799	0.63–1.014	0.747	0.575–0.972
Prostate cancer										
ACEI	No	581,622	298	0.51	1.000 (reference)		1.000 (reference)		1.000 (reference)	
	Yes	43,328	28	0.65	1.165	0.789–1.719	0.791	0.534–1.71	0.764	0.513–1.137
ARB	No	134,514	82	0.61	1.000 (reference)		1.000 (reference)		1.000 (reference)	
	Yes	490,436	244	0.50	0.84	0.654–1.08	0.954	0.739–1.232	0.974	0.741–1.279
BB	No	495,889	248	0.50	1.000 (reference)		1.000 (reference)		1.000 (reference)	
	Yes	129,060	78	0.60	1.144	0.888–1.474	1.057	0.82–1.362	1.099	0.848–1.425
CCB	No	162,608	88	0.54	1.000 (reference)		1.000 (reference)		1.000 (reference)	
	Yes	462,342	238	0.51	0.917	0.717–1.172	0.813	0.635–1.041	0.85	0.657–1.1
Diuretics	No	318,213	173	0.54	1.000 (reference)		1.000 (reference)		1.000 (reference)	
	Yes	306,737	153	0.50	0.883	0.712–1.096	0.865	0.697–1.074	0.892	0.707–1.125
Pancreatic cancer										
ACEI	No	582,672	90	0.15	1.000 (reference)		1.000 (reference)		1.000 (reference)	
	Yes	43,445	7	0.16	0.959	0.446–2.064	0.827	0.383–1.787	0.893	0.421–1.897
ARB	No	134,810	26	0.19	1.000 (reference)		1.000 (reference)		1.000 (reference)	
	Yes	491,306	71	0.14	0.774	0.493–1.226	0.865	0.55–1.361	0.771	0.474–1.254
BB	No	496,794	86	0.17	1.000 (reference)		1.000 (reference)		1.000 (reference)	
	Yes	129,323	11	0.09	0.459	0.246–0.857	0.448	0.24–0.837	0.447	0.243–0.822
CCB	No	162,924	19	0.12	1.000 (reference)		1.000 (reference)		1.000 (reference)	
	Yes	463,193	78	0.17	1.383	0.84–2.277	1.316	0.799–2.167	1.326	0.797–2.206
Diuretics	No	318,887	47	0.15	1.000 (reference)		1.000 (reference)		1.000 (reference)	
	Yes	307,230	50	0.16	1.057	0.712–1.571	1.039	0.699–1.545	1.189	0.774–1.827
Kidney cancer										
ACEI	No	582,302	110	0.19	1.000 (reference)		1.000 (reference)		1.000 (reference)	
	Yes	43,414	10	0.23	1.241	0.656–2.347	1.123	0.594–2.124	1.102	0.577–2.105
ARB	No	134,721	28	0.21	1.000 (reference)		1.000 (reference)		1.000 (reference)	
	Yes	490,995	92	0.19	0.909	0.595–1.389	0.881	0.576–1.347	0.846	0.534–1.339
BB	No	496,511	93	0.19	1.000 (reference)		1.000 (reference)		1.000 (reference)	
	Yes	129,205	27	0.21	1.135	0.737–1.749	1.111	0.721–1.712	1.084	0.692–1.698
CCB	No	162,840	25	0.15	1.000 (reference)		1.000 (reference)		1.000 (reference)	
	Yes	462,876	95	0.21	1.353	0.867–2.11	1.309	0.838–2.045	1.328	0.845–2.088
Diuretics	No	318,716	59	0.19	1.000 (reference)		1.000 (reference)		1.000 (reference)	
	Yes	307,000	61	0.20	1.089	0.762–1.556	1.099	0.769–1.57	1.127	0.766–1.658

ACEI, angiotensin converting enzyme inhibitor; ARB, angiotensin receptor blocker; BB, beta blocker; CCB, calcium channel blocker; HR, hazard ratio; CI, confidence interval.

## Data Availability

Due to the nature of this research, participants of this study did not agree for their data to be shared publicly, so Appendix A are not available.

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
