# Peer review of "Antihypertensive Drugs and the Risk of Cancer: A Nationwide Cohort Study"

_jcm, 2021, doi:10.3390/jcm10040771_

Round 1

Reviewer 1 Report

Cho et al. evaluated impact of different antihypertensive drug use on the risk of incident cancer in Korean population-based retrospective cohort study. As a result it seems that the use of angiotensin II receptor blockers decrease the risk of incident cancer. This is interesting study, however, some points need to be clarified.

The most major issue in this study is that non-users of any antihypertensive drugs were not included in the analysis. You concluded that use of overall antihypertensive drugs was not associated with increased cancer incidence. However, you have compared users of different antihypertensive drugs on users of other class of antihypertensive drugs. Such a conclusion cannot be done without comparing users of antihypertensive drugs to non-users of antihypertensive drug. Thus conclusion needs to be clarified or conduct a new analysis where actual non-users are used as a reference group.

The study showed that the use of ARBs seems to decrease risk of incident cancer compared to users of other antihypertensive drug. It would also be interesting to see the results if comparison were made with non-users of any antihypertensive drug.

You state in your limitations that ARB users were more prevalent compared to other antiHT drug users. Please, clarify this. If ARB users differ from other antihypertensive drug users, it needs more discussion on how and how it might bias the results.

It is not clear did you use antihypertensive drug use as a time-dependent variable in the analysis? If not risk of time-immortal bias should be discussed in the limitations especially if new analysis are made with non-users of any antihypertensive drugs.

SBP level values were reported from the time of the hypertension diagnosis. Mean values (around 140 mmHg) seem to be a bit low for subjects without drug treatment and still have the diagnosis of hypertension. Furthermore, in Supplemental Figure 1 subgroup analysis with different SBP levels among ARB users and non-users are made. Based on the figure only 14% ARB non-users and 18.5% ARB users have SBP over 140 mmHg. This raises the question whether SBP levels are collected after drug treatment has already implemented? 

In general, hypertensive patients commonly need more than one class of antihypertensive drugs in order to reach to normal values of blood pressure. Did you evaluate amount of such subjects in your data? One possible sub-analysis could be evaluating risk of incident cancer among subjects which have simultaneous use of two or more different class of antihypertensive drugs e.g. ARBs and diuretics.

Discussion on impact of hypertension on risk of incident cancer is missing.

English editing would make the text clearer and easier to read

Spelling errors: Line 160: 1000 person-years?, Line 266: the renin-aldosterone-angiotensin system --> change renin-angiotensin-aldosterone system. Later on this is named as renin-angiotensin system (line 276). Use similar naming throughout the text.

Author Response

Responses to the reviewers’ comments

Reviewer 1

Cho et al. evaluated impact of different antihypertensive drug use on the risk of incident cancer in Korean population-based retrospective cohort study. As a result it seems that the use of angiotensin II receptor blockers decrease the risk of incident cancer. This is interesting study, however, some points need to be clarified.

Comment #1. The most major issue in this study is that non-users of any antihypertensive drugs were not included in the analysis. You concluded that use of overall antihypertensive drugs was not associated with increased cancer incidence. However, you have compared users of different antihypertensive drugs on users of other class of antihypertensive drugs. Such a conclusion cannot be done without comparing users of antihypertensive drugs to non-users of antihypertensive drug. Thus conclusion needs to be clarified or conduct a new analysis where actual non-users are used as a reference group.

Response: We appreciate the reviewer’s invaluable comment. However, we have compared users of different antihypertensive drugs on users of other class of antihypertensive drugs, as the purpose of the current study was to invest the risk of cancer among patients who had taken antihypertensive drugs and to compare the risks among the different antihypertensive drugs, and to guide the choice of antihypertensive drugs for long-term usage as a perspective of cancer incident risk. However, we agree that it would be more proper to emphasize that the decreased cancer incidence risk in ARB users is expressed relative to users of other anti-hypertensive drugs and not compared to the general population.

Therefore, we modified the conclusion as follows; “The users of common antihypertensive medications were not associated with increased risk of cancer overall compared to users of other classes of antihypertensive drugs. ARB was independently associated with a decreased risk of cancer overall compared to other antihypertensive drugs.”

Comment #2. The study showed that the use of ARBs seems to decrease risk of incident cancer compared to users of other antihypertensive drug. It would also be interesting to see the results if comparison were made with non-users of any antihypertensive drug.

Response: We agree with the reviewer’s comment. It would be ideal to compare cancer incidence of patients who had taken only one class of antihypertensive drugs to that of non-users of any antihypertensive drugs. However, as many of the hypertensive patients have been taken more than 2 classes of antihypertensive drugs, the number of patients who have been taken only one class of antihypertensive drugs was too small to conduct an analysis, which led to the current research protocol. In response to reviewer’s comment, we have added this issue in the limitation section, as follows; “Fourth, the study population of this study included only antihypertensive drug users; decreased cancer incidence risk in ARB users is expressed relative to users of other antihypertensive drugs and not compared to the general population.” (page 17, limitation section).

Comment #3. You state in your limitations that ARB users were more prevalent compared to other antiHT drug users. Please, clarify this. If ARB users differ from other antihypertensive drug users, it needs more discussion on how and how it might bias the results.

Response: During the study period, the percentage of antihypertensive drugs used in the study population was 79%, 72%, 47%, 24%, and 6% for ARB, CCB, diuretics, BB, and ACEI, respectively. ARB users not differ from other antihypertensive drug users, while the number of ACEI users was very small. This small number of ACEI users might cause the bias, and we described this fact in the manuscript, as follows; “Fifth, the prevalence of ACEI users was very low compared with those taking other antihypertensive drugs, which might have affected statistical results.” (page 16, limitation section).

Comment #4. It is not clear did you use antihypertensive drug use as a time-dependent variable in the analysis? If not risk of time-immortal bias should be discussed in the limitations especially if new analysis are made with non-users of any antihypertensive drugs.

Response: We added the possibility of time-immortal bias in the limitation section, as follows; “since we did not incorporate the antihypertensive drug usage as a time-dependent variable, immortal time bias can occur, although individuals enrolled in this study were taking at least one antihypertensive drug.” (page 17, limitation section)

Comment #5. SBP level values were reported from the time of the hypertension diagnosis. Mean values (around 140 mmHg) seem to be a bit low for subjects without drug treatment and still have the diagnosis of hypertension. Furthermore, in Supplemental Figure 1 subgroup analysis with different SBP levels among ARB users and non-users are made. Based on the figure only 14% ARB non-users and 18.5% ARB users have SBP over 140 mmHg. This raises the question whether SBP levels are collected after drug treatment has already implemented? 

Response: For SBP levels, we used the most recent NHIS health screening database from the time of hypertension diagnosis. Therefore, as the reviewer mentioned, the SBP level might be measured after the diagnosis and treatment of hypertension in considerate amount of population. This might result in low mean SBP values.  

Comment #6. In general, hypertensive patients commonly need more than one class of antihypertensive drugs in order to reach to normal values of blood pressure. Did you evaluate amount of such subjects in your data? One possible sub-analysis could be evaluating risk of incident cancer among subjects which have simultaneous use of two or more different class of antihypertensive drugs e.g. ARBs and diuretics.

Response: We appreciate the reviewer’s invaluable comment. Sub-analysis of evaluating the risk of incident cancer among subjects who have simultaneous use of two or more different classes of antihypertensive drugs would be very beneficial, although this is beyond our goal of this study. However, future studies are warranted, and we added this comment for the future direction of the studies, as follows; “Additional analysis evaluating the risk of incident cancer among subjects with simultaneous use of two or more different classes of antihypertensive drugs would also provide useful insight.” (page 17, limitation section).

Comment #7. Discussion on impact of hypertension on risk of incident cancer is missing.

Response: We added the discussion on the impact of hypertension on the risk of incident cancer, as the reviewer suggested (page 15, discussion section).

Comment #8. English editing would make the text clearer and easier to read

Response: English editing was performed through the manuscript, as the reviewer suggested.

Comment #9. Spelling errors: Line 160: 1000 person-years?, Line 266: the renin-aldosterone-angiotensin system --> change renin-angiotensin-aldosterone system. Later on this is named as renin-angiotensin system (line 276). Use similar naming throughout the text.

Response: We changed all spelling errors, as the reviewer suggested.

Finally, we once again would like to thank reviewers for his/her excellent comments that significantly enhanced the quality of the manuscript.

Reviewer 2 Report

This is a conveniently concise article on a timely topic. Some modifications could still be incorporated, and some points of attention can increase the quality and clarity of the manuscript.

Abstract

  • Consider emphasizing that the decreased cancer incidence risk in ARB users is expressed relative to users of other anti-hypertensive drugs and not compared to the general population.

Introduction

  • Rationale weakly described, there are more previous studies assessing this correlation, better to refer rather than pretend it is the group’s own hypothesis.
  • It would be of added value to have more background included concerning the possible anti-cancer effects of antihypertensive drugs, e.g. by addition of mechanistic evidence/studies that provide a biological rationale for the possible protective phenomenon. This could alternatively be discussed in the discussion more elaboratively.
  • concerns were raised that the drugs might increase the risk of cancer => better keep it neutral, also protective effects have been described previously
  • Line 83: consider adding the names of the different organizations that established these hypertension guidelines.
  • Line 96-99: The sentence “since the introduction of ARBs in the 1990s, which are the most recently developed antihypertensive drug class, several decades have passed, and now we have enough elapsed time and clinical data to analyze the long-term effects of antihypertensive drugs on incident cancer”: this is true, but the current analysis does not have decades of follow-up (median FU: 8.6 years). Therefore, consider rewriting this sentence.

Methods

  • First paragraph on data access: better move down and start with description of methods, study population etcetera
  • NHIS: data on all citizens? Needs to be clear to assess risk of selection bias
  • 70k patients remaining from 537k patients with hypertension: especially the prevalent users, why excluded?. Also the high number of patients without medication or using mediction only<1 year is striking. This raises questions about accountability of the population
  • Please specify the reasons underlying the exclusion of patients that used antihypertensive drugs at baseline or that used antihypertensive drugs < 1 year from the analysis.

Results

  • Taken together, there were 157039 users of antihypertensive drugs divided over the study population of 70,549. This would mean an average of more than 2 antihypertensive drugs for each included patient. Were patients with more types of anti-hypertensive use more or less vulnerable to develop cancer? And could combination therapy trouble the results observed in the ARB group? Please remark on this.
  • 4% of all included patients (70,549) developed cancer during the median follow-up of 8.6 years. Do we know the absolute cancer incidence in the ARB group? Adding this could provide more clarity, and possible robustness, to the data.

Results

How were changes in medication handled?

Very low number of ACE inhibitors, are ARBs first choice in Korea?

How were analyses performed when several antihypertensives were used? And were classes compared with each other?

Use of 5-6 or even 9 years is very short, same for the lag time

Analyses per type of cancer: caveat multiple testing

Discussion

  • Please include that the reference population used consisted of patients that used other antihypertensives.
  • Line 245: next to median follow-up period, consider adding the median drug exposure durations to better clarify the findings.
  • In general, this is a high-income and relatively young population with a short period of antihypertensive drug use. Please elaborate whether the results from this study would be expected to be more or less solid in a more generalizable population with longer median duration of antihypertensive drug use.
  • The effect of antihypertensive drugs on SBP does not seem to influence cancer risk. However, in another Korean study, the height of SBP seem to contribute to RCC incidence risk (Kim et al. 2020 Hypertension, PMID: 32336229, PMCID: PMC7682799 DOI: 10.116/HYPERTENSIONAHA.120.14820. Consider discussing this possible discrepancy between results.
  • Add considering the limitation that control on drug adherence was lacking
  • Please specify, if known, what proportion used their indicated antihypertensive therapy during the entire follow-up period.

Tables and figures

  • Table 1 indicates income deciles and indicates 20 parts. However, a decile consists of 10 parts in total.
  • Table 2: consider changing “incidence” into “cancer incidence” to add to the clarity.

Author Response

Responses to the reviewers’ comments

Reviewer 2

This is a conveniently concise article on a timely topic. Some modifications could still be incorporated, and some points of attention can increase the quality and clarity of the manuscript. 

Abstract

Comment #1. Consider emphasizing that the decreased cancer incidence risk in ARB users is expressed relative to users of other anti-hypertensive drugs and not compared to the general population.

Response: We modified the conclusion as follows; “The users of common antihypertensive medications were not associated with increased risk of cancer overall compared to users of other classes of antihypertensive drugs. ARB was independently associated with a decreased risk of cancer overall compared to other antihypertensive drugs.”

Introduction 

Comment #2. Rationale weakly described, there are more previous studies assessing this correlation, better to refer rather than pretend it is the group’s own hypothesis.

Response: We modified the introduction by describing more previous studies, as the reviewer suggested (page 7-8, introduction section).  

Comment #3. It would be of added value to have more background included concerning the possible anti-cancer effects of antihypertensive drugs, e.g. by addition of mechanistic evidence/studies that provide a biological rationale for the possible protective phenomenon. This could alternatively be discussed in the discussion more elaboratively.

Response: We modified the introduction, as the reviewer suggested.

Comment #4. concerns were raised that the drugs might increase the risk of cancer => better keep it neutral, also protective effects have been described previously

Response: In response to the reviewer’s comments, we removed the sentence.

Comment #5. Line 83: consider adding the names of the different organizations that established these hypertension guidelines.

Response: We added the name of the organizations that established this hypertension guideline, as follows; “The 2017 American College of Cardiology/American Heart Association guidelines for high blood pressure for adults recommend diuretics, angiotensin-converting enzyme inhibitors (ACEIs), angiotensin receptor blockers (ARBs), and calcium channel blockers (CCBs) as primary agents and beta blockers (BBs) and alpha blockers as secondary agents for hypertension treatment in cases without compelling indications such as heart failure, kidney disease, or angina..

Comment #6. Line 96-99: The sentence “since the introduction of ARBs in the 1990s, which are the most recently developed antihypertensive drug class, several decades have passed, and now we have enough elapsed time and clinical data to analyze the long-term effects of antihypertensive drugs on incident cancer”: this is true, but the current analysis does not have decades of follow-up (median FU: 8.6 years). Therefore, consider rewriting this sentence.

Response: We rewrote the sentence, as follow; ARBs, the most recently developed antihypertensive drug class, were introduced in the 1990s,15 and the long-term use of ARB in relation to incident cancer still needs further investigation. The potential for cancer risk of ARB was first raised in the candesartan trial,16 which followed multiple studies with conflicting results.17-21 ARBs were associated with an increased risk of cancer development in a meta-analysis,17 while two other subsequent meta-analyses did not find excess cancer risk among ARB users compared with scontrol.18, 19 Furthermore, several mechanistic pieces of evidence provide a biological rationale for the possible antitumor effect of ARBs via angiotensin-type 2 receptor (AT2R) stimulation.22-24. (page7-8, introduction section)

Methods

Comment #7. First paragraph on data access: better move down and start with description of methods, study population etcetera

Response: We moved down the paragraph, as the reviewer suggested.

Comment #8. NHIS: data on all citizens? Needs to be clear to assess risk of selection bias

Response: We clarified it as follows (page 8); “Patients who were diagnosed with essential hypertension were identified with International Classification of Diseases, 10th revision (ICD-10) codes I10-13 between January 1, 2005 and December 31, 2012, from all the National Health Insurance System (NHIS) database in Korea.” In addition, we added this issue in Method section at page 10 as follows: “National Health Insurance Service is a single insurance provider in Korea and covers 97% of the Korean population.”

Comment #9. 70k patients remaining from 537k patients with hypertension: especially the prevalent users, why excluded?

Also the high number of patients without medication or using mediction only<1 year is striking. This raises questions about accountability of the population

Response: For prevalent antihypertensive drug users, we cannot assess the duration of the drug use, and more importantly, it was not clear whether patients had taken other antihypertensive medications before the enrollment, which could mislead results. Moreover, it has reported that prevalent users are "survivors" of the early period of pharmacotherapy, which can introduce substantial bias if risk varies with time (Am J of epidemiology 2003:158;915~920). Therefore, we excluded the prevalent users and only included new drug users for the analysis. For patients using medication <1year was also excluded, because the cumulative effect of drug might not be enough to cause cancer.

The diagnosis in NHIS primarily serves the purpose of administrative billing and was not originally intended to perform statistical analysis, which might cause a high number of patients without medication or using medication only<1 year. However, the data on the prescription of antihypertensive drugs and cancer diagnoses were very reliable. We described this fact in the manuscript as a limitation, as follows; “Second, the diagnosis in NHIS primarily serves the purpose of administrative billing and was not originally intended to perform statistical analysis. However, data on the prescription of antihypertensive drugs and cancer diagnoses were very reliable. Third, we did not have data regarding the dosage of antihypertensive drugs, which did not permit the analysis of the association between cumulative dose and cancer incidence, which might have strengthened our hypothesis.” (page 17, limitation section).

Comment #10. Please specify the reasons underlying the exclusion of patients that used antihypertensive drugs at baseline or that used antihypertensive drugs < 1 year from the analysis.

Response: For prevalent antihypertensive drug users, we cannot assess the duration of the drug use, and more importantly, it is not clear whether patients had taken other antihypertensive drugs before the enrollment, which could mislead the result. Therefore, we excluded the prevalent users. We added this fact to the manuscript, as follows; “As the duration of drug use and whether the patient had taken other antihypertensive drugs before the index period could not be assessed, we excluded prevalent users and those who used antihypertensive drugs at baseline.” (page8).

Results

Comment #11. Taken together, there were 157039 users of antihypertensive drugs divided over the study population of 70,549. This would mean an average of more than 2 antihypertensive drugs for each included patient. Were patients with more types of anti-hypertensive use more or less vulnerable to develop cancer? And could combination therapy trouble the results observed in the ARB group? Please remark on this.

Response: We appreciate the reviewer’s invaluable comment. Sub-analysis of evaluating the risk of incident cancer among subjects which have simultaneous use of two or more different class of antihypertensive drugs would be very beneficial, although this is beyond our goal of the current study. However, future studies are warranted, and therefore we added this comment for future direction of the studies, as follows; “Additional analysis evaluating the risk of incident cancer among subjects with simultaneous use of two or more different classes of antihypertensive drugs would also provide useful insight.” (page 16, limitation section).

Comment #12. 4% of all included patients (70,549) developed cancer during the median follow-up of 8.6 years. Do we know the absolute cancer incidence in the ARB group? Adding this could provide more clarity, and possible robustness, to the data.

Response: The number of overall cancers is 3,294 patients in total of 55,645 patients in the ARB group. The result is presented in Table1, and we added the absolute cancer incidence in the manuscript (page13, result section). 

Comment #13. How were changes in medication handled?

Response: Participants who were prescribed antihypertensive medications for at least 1 year were defined as medication users. Non-users were subjects who had never used the medication. Patients who were prescribed antihypertensive medications for at least 1 year for more than 2 antihypertensive drugs, he was coded as a user for all drugs.

Comment #14. Very low number of ACE inhibitors, are ARBs first choice in Korea?

Conclusion: ACEI has been used less compared to ARBs in Korea, which might be caused by the relatively high prevalence of cough with ACEI usage in Asians. According to the Korea hypertension fact sheet 2018(Clinical Hypertension,Vol.24:13, 2018), ARB was prescribed in 47.3% of patients, while ACEI was prescribed in 1.4% of patients, as a monotherapy for hypertension in Korea.

Comment #15. How were analyses performed when several antihypertensives were used? And were classes compared with each other?

Response: Participants who were prescribed antihypertensive medications for at least 1 year were defined as medication users. Non-users were subjects who had never used the medication. Patients who were prescribed antihypertensive medications for at least 1 year for more than 2 antihypertensive drugs, he was coded as a user for all drugs.

Comment #16. Use of 5-6 or even 9 years is very short, same for the lag time

Response: The short duration would be the limitation of the current study, and we added this fact in the manuscript, as follows; “In general, this cohort represented a high-income and relatively young Korean population with a short period of antihypertensive drug use. Therefore, it may not be possible to generalize these results to a broader population with a longer median duration of antihypertensive drug use.” (page 16, limitation section).

Comment #17. Analyses per type of cancer: caveat multiple testing

Response: The primary outcome was cancer incident overall, and we agree that analyses per type of cancer caveat multiple testing. We added this fact in the manuscript, as follows; “From site-specific analysis, we speculated that the reduced cancer risk associated with ARB use is driven by a decrease in hepatic cancer and gastric cancer, although our analysis per type of cancer should be interpreted with caution and warrant further investigation.” (page 15, discussion section).  

Discussion

Comment #18. Please include that the reference population used consisted of patients that used other antihypertensives.

Response: We modified the first part of the discussion, as follows; “The principal finding of the current study was that users of common antihypertensive medications were not associated with increased risk of cancer overall compared to users of other hypertensive patients. Among 5 common antihypertensive drugs including ACEI, BB, CCB, ARB, and diuretics, ARB was independently associated with a decreased risk of cancer overall compared to other antihypertensive drugs, and the risk of incident cancer overall was decreased further with a longer duration of ARB use.”

Comment #19. Line 245: next to median follow-up period, consider adding the median drug exposure durations to better clarify the findings.

Response: We added the median drug exposure duration, as the reviewer suggested (page 11).

Comment #20. In general, this is a high-income and relatively young population with a short period of antihypertensive drug use. Please elaborate whether the results from this study would be expected to be more or less solid in a more generalizable population with longer median duration of antihypertensive drug use.

Response: We added the description, as follows; “In general, this is a high-income and relatively young Korean population with a short period of antihypertensive drug use. Therefore, the application of the results from this study to a more generalizable population with a longer median duration of antihypertensive drug use needs further investigation (page 17, limitation section).”

Comment #21. The effect of antihypertensive drugs on SBP does not seem to influence cancer risk. However, in another Korean study, the height of SBP seem to contribute to RCC incidence risk (Kim et al. 2020 Hypertension, PMID: 32336229, PMCID: PMC7682799 DOI: 10.116/HYPERTENSIONAHA.120.14820. Consider discussing this possible discrepancy between results.

Response: The current study population was those with antihypertensive drugs and their SBP was relatively well-controlled, which might blunt the impact of SBP on RCC incidence. We added discussion as follows; “Interestingly, the risk of kidney cancer significantly increased with higher blood pressure, in a dose-dependent manner, even after adjusting for antihypertensive medication use in the study of the Korean cohort.28 Our study population consisted of those with mean SBP less than 140 mmHg, suggesting well-controlled status of blood pressure, and this might suggest the impact of blood pressure rather than antihypertensive drugs on cancer incidence, which needs further investigation” (page 15-16, discussion section).

Comment #22. Add considering the limitation that control on drug adherence was lacking

Response: We added the limitation that control on drug adherence was lacking, as the reviewer suggested (page 17, limitation section).

Comment #23. Please specify, if known, what proportion used their indicated antihypertensive therapy during the entire follow-up period.

Response: Unfortunately, we do not have the data regarding the accurate proportion used their indicated antihypertensive therapy during the entire follow-up. However, mean drug exposure duration is between 3.3 ~ 5.5 years and mean follow-up years is between 8.5 ~10.7 years, and therefore, the duration of drug continuation would be enough to investigate the impact of antihypertensive drugs.

Tables and figures

Comment #24. Table 1 indicates income deciles and indicates 20 parts. However, a decile consists of 10 parts in total.

Response: We corrected the error.

Comment #25. Table 2: consider changing “incidence” into “cancer incidence” to add to the clarity.

Response: We changed it, as the reviewer suggested.

Finally, we once again would like to thank reviewers for his/her excellent comments that significantly enhanced the quality of the manuscript.

Round 2

Reviewer 1 Report

Thank you for the responses.

Author Response

Thank you for your e-mail regarding the possible publication of our manuscript in Journal of Clinical Medicine.

We appreciate the reviewer and editor’s invaluable comment. However, unfortunately, it is impossible to do the recommended additional analysis within 2 days. We do not have the Korean National Health Insurance System data in an easily accessible location. We have to obtain the analysis permission from the center again, visit the data analytic lab located in a city within hours, and analyze the data which takes several days. Even worse, this whole system is limited and needs more months due to the current pandemic COVID-19 situation in Korea.

Therefore, very unfortunately, we added the limitation as follows; “Fourth, the study population of this study included only antihypertensive drug users; decreased cancer incidence risk in ARB users is expressed relative to users of other antihypertensive drugs and not compared to the general population. Further analysis of a comparison of users of antihypertensive drugs to those not using antihypertensive drugs would be needed to reduce bias.” (page8, lines 336-340)

If the editor still requests further analysis necessarily, we kindly ask for a few more months to do the process.

We thank you again for your consideration and hope that the revised manuscript would meet your approval for publication.